# Review and Meta-Analysis of the Evidence for Choosing between Specific Pyrethroids for Programmatic Purposes

**DOI:** 10.3390/insects12090826

**Published:** 2021-09-14

**Authors:** Natalie Lissenden, Mara D. Kont, John Essandoh, Hanafy M. Ismail, Thomas S. Churcher, Ben Lambert, Audrey Lenhart, Philip J. McCall, Catherine L. Moyes, Mark J. I. Paine, Giorgio Praulins, David Weetman, Rosemary S. Lees

**Affiliations:** 1Department of Vector Biology, Liverpool School of Tropical Medicine, Liverpool L3 5QA, UK; Natalie.Lissenden@lstmed.ac.uk (N.L.); John.Essandoh@lstmed.ac.uk (J.E.); Hanafy.Ismail@lstmed.ac.uk (H.M.I.); Philip.McCall@lstmed.ac.uk (P.J.M.); Mark.Paine@lstmed.ac.uk (M.J.I.P.); Giorgio.Praulins@lstmed.ac.uk (G.P.); David.Weetman@lstmed.ac.uk (D.W.); 2MRC Centre for Global Infectious Disease Analysis, School of Public Health, Imperial College London, London SW7 2BX, UK; mara.kont17@imperial.ac.uk (M.D.K.); thomas.churcher@imperial.ac.uk (T.S.C.); ben.c.lambert@gmail.com (B.L.); 3U.S. Centers for Disease Control and Prevention, Entomology Branch, Division of Parasitic Diseases and Malaria, Atlanta, GA 30329, USA; ajl8@cdc.gov; 4Big Data Institute, University of Oxford, Oxford OX3 7LF, UK; catherinemoyes@gmail.com

**Keywords:** pyrethroid, pyrethroid resistance, insecticide resistance, insecticide resistance management, vector control, malaria, malaria control, mosquito, *Anopheles*

## Abstract

**Simple Summary:**

A group of insecticides, called pyrethroids, are the main strategy for controlling the mosquito vectors of malaria. Pyrethroids are used in all insecticide-treated bednets, and many indoor residual spray programmes (in which insecticides are sprayed on the interior walls of houses). There are different types of pyrethroids within the class (e.g., deltamethrin and permethrin). Across the world, mosquitoes are showing signs of resistance to the pyrethroids, such as reduced mortality following contact. However, it is unclear if this resistance is uniform across the pyrethroid class (i.e., if a mosquito is resistant to deltamethrin, whether it is resistant to permethrin at the same level). In addition, it is not known if switching between different pyrethroids can be used to effectively maintain mosquito control when resistance to a single pyrethroid has been detected. This review examined the evidence from molecular studies, resistance testing from laboratory and field data, and mosquito behavioural assays to answer these questions. The evidence suggested that in areas where pyrethroid resistance exists, different mortality seen between the pyrethroids is not necessarily indicative of an operationally relevant difference in control performance, and there is no reason to rotate between common pyrethroids (i.e., deltamethrin, permethrin, and alpha-cypermethrin) as an insecticide resistance management strategy.

**Abstract:**

Pyrethroid resistance is widespread in malaria vectors. However, differential mortality in discriminating dose assays to different pyrethroids is often observed in wild populations. When this occurs, it is unclear if this differential mortality should be interpreted as an indication of differential levels of susceptibility within the pyrethroid class, and if so, if countries should consider selecting one specific pyrethroid for programmatic use over another. A review of evidence from molecular studies, resistance testing with laboratory colonies and wild populations, and mosquito behavioural assays were conducted to answer these questions. Evidence suggested that in areas where pyrethroid resistance exists, different results in insecticide susceptibility assays with specific pyrethroids currently in common use (deltamethrin, permethrin, α-cypermethrin, and λ-cyhalothrin) are not necessarily indicative of an operationally relevant difference in potential performance. Consequently, it is not advisable to use rotation between these pyrethroids as an insecticide-resistance management strategy. Less commonly used pyrethroids (bifenthrin and etofenprox) may have sufficiently different modes of action, though further work is needed to examine how this may apply to insecticide resistance management.

## 1. Introduction

Pyrethroids are present in all WHO-prequalified insecticide-treated nets (ITNs), and are also used for indoor residual spraying (IRS) [1]. Pyrethroid resistance is widespread in malaria vectors [2,3], and differential mortality in discriminating dose bioassays between pyrethroids is often observed during susceptibility monitoring in lab strains and in wild populations. There is uncertainty about whether current methods for monitoring insecticide resistance can reliably identify moderately different levels of resistance within the pyrethroid class. When differential mortality is observed in discriminating dose bioassays, it is unclear if this should be interpreted as an indication of differential levels of susceptibility within the pyrethroid class, or if this could arise due to inherent variability in bioassay results or differently calibrated discriminating doses within the pyrethroid class. Considering this, when differential susceptibility is observed, there is a question regarding whether countries can use targeted or preferential use of specific pyrethroid insecticides as an effective resistance management strategy. This is important, as maintaining the efficacy of pyrethroids is vital to malaria control while we wait for novel active ingredients (AIs) with new modes of action (MoA) to be developed. To address these questions, this review examined evidence from molecular studies, insecticide resistance patterns and bioassay results from laboratory colonies and field populations, and lessons from behavioural assays.

## 2. Do Discriminating Doses Accurately Detect Resistance in Different Pyrethroids?

Current methods for monitoring insecticide resistance are based on classifying phenotypic resistance, which is typically measured using standardised tests, such as WHO susceptibility bioassays [4] and CDC bottle assays [5]. These tests expose mosquito populations (wild-collected females or those reared from collected larvae) to predefined “discriminating doses” (DDs) of an insecticide, and record mosquito knockdown and mortality at defined times postexposure. A DD is defined by WHO [4] as “a concentration of an insecticide that, in a standard period of exposure, is used to discriminate the proportions of susceptible and resistant phenotypes in a sample of a mosquito population”. It is calculated by establishing a dose response in susceptible mosquitoes, and then calculating either “twice the lowest concentration that gave systematically 100% mortality (i.e., LC_100_)” or “twice the LC_99_ values” estimated from this baseline susceptibility testing. Under- or overestimation of discriminating doses could have an impact on the accurate detection of insecticide resistance in wild populations, and misclassification of lab strains.

In 2016, following increasing evidence of the limitations of discriminating dose assays [6,7], the WHO updated their monitoring guidance to include additional testing of resistant populations at 5× and 10× DDs to provide further information on the intensity or “strength” of phenotypic resistance [4].

The current WHO DDs for *Anopheles* for deltamethrin (0.05%) and permethrin (0.75%) were established, along with other pyrethroids, through an international multicentre study in 1998 [8]. Some other pyrethroids, including α-cypermethrin, were not included in this original study; the α-cypermethrin recommended DD (0.05%) is tentative, and currently under validation by the WHO in a new multicentre study [9]. The 1998 study exposed known insecticide-“susceptible” strains of *Anopheles albimanus*, *Anopheles gambiae*, and *Anopheles stephensi* to up to five different concentrations of each insecticide using WHO tube bioassays. Mortality postexposure was then analysed using Probit regression to establish a single lethal dose for all *Anopheles* species for each compound, which was doubled to give the DDs that are still recommended today.

In this review, we reanalysed the publicly available data from the 1998 study to establish if the recommended DDs are suitable and comparable. Using these data, we were unable to establish the LC_100_ for permethrin, since in one centre some mosquitoes survived the highest concentration tested (Mali, *An. gambiae*, Mopti strain, 1% permethrin, 99.2% mortality). For deltamethrin, data were incomplete (0.1% deltamethrin killed 100% of the exposed mosquitoes, but not all centres tested the 0.1% concentration against all strains).

We then used Probit analysis (PoloJR program within PoloSuite, Version 2.1) to obtain LD_99_ values. The DDs calculated from this when all the data were pooled were 0.1% for deltamethrin and 1.46% for permethrin (Figure 1, Table 1), around double the final DDs recommended by the 1998 multicentre study [8]. The concentrations tested in the original study did not produce a full range of mortality (i.e., 0–100%), which resulted in poorly fitting dose-response curves. Poorly fitting dose-response curves were also observed when strains and species were pooled separately (Appendix A). In several cases, our Probit analysis could not calculate the lethal concentration for individual sites/strains or estimate meaningful confidence intervals around the LC values (Appendix A). Robust dose-response relationships were not observed, and in some study sites, mortality was never below 80% in the strain tested. The original selection of DDs was thus not well supported by the data.

Most centres within the study appeared to have diverged from the common protocol in terms of sample size and replicates tested. In the original protocol, 2–3 replicates of 100 mosquitoes (200–300 mosquitoes) should have been tested per insecticide concentration. However, at some study sites, *n* < 25 mosquitoes per concentration were tested. Mosquitoes were tested at 1—3 days old, whereas current guidelines state 3–5-day-old mosquitoes should be used [4]. This may have influenced results, particularly in 1-day-olds, as their cuticles may not have hardened [10].

In the original report, there are a lack of raw data, and it is not clear which data were used in proposing the final doses. The report states that more weight was given to studies in which mortalities were clustered around similar values; however, which analyses were weighted or the methodology for weighting was not specified. It is therefore not possible to establish why the WHO-recommended DDs differed from the ones calculated in this review. Given the unclear rationale for the DDs recommended for permethrin and deltamethrin, and that the data used to calculate them were unclear, their comparability is questionable. The DDs of these pyrethroids were not calibrated against one another, and the assay was not designed to compare compounds, but to monitor for resistance to each independently. This was a challenge when trying to draw reliable conclusions about the relative efficacy of, or resistance to, these two pyrethroids based on data collected using these DDs.

## 3. Is there Molecular Evidence for Differential Resistance among Members of the Pyrethroid Insecticide Class?

Molecular studies indicate that structurally diverse pyrethroids such as tefluthrin, transfluthrin, bifenthrin, and etofenprox, which lack the common structural moiety of most pyrethroids, may interact differently with the common resistance mechanisms found in insect populations [11,12,13] (Figure 2). To assess cross-resistance within the pyrethroids in terms of their interactions with key cytochrome P450 enzymes (hereafter P450s) and resistance in vector populations, P450 functional activity data with pyrethroids were compared with field mortality data [12]. Figure 3 shows the relationships among pyrethroids in terms of their binding affinity to and depletion by key P450 enzymes known to confer metabolic pyrethroid resistance in *Anopheles gambiae* (s.l.) in comparison to mortality data among pyrethroids. Bifenthrin diverges from the pyrethroids commonly used in malaria vector control in terms of binding affinity to, and depletion by, P450s from African *Anopheles* species, while etofenprox diverges from the other pyrethroids in terms of binding affinity to these P450s, but not depletion by them [12]. When these relative differences found by molecular studies were compared to relative differences in the prevalence of resistance to each pyrethroid within African malaria vector populations, the potential divergence of etofenprox was observed in both the molecular studies and the field studies, but bifenthrin has not yet been tested in field studies of African malaria vectors (Figure 3).

## 4. What Intrinsic Variability Do We See from Dose-Response Assays in the Lab?

Discriminating dose bioassays are routinely used to detect and monitor insecticide resistance in mosquito populations. When conducted in well-controlled lab settings, factors such as temperature, humidity, and mosquito rearing are standardised to minimise their effects on mosquito mortality. Examining repeated measurements taken in these settings, with as many variables as possible controlled, allowed us to investigate what intrinsic variability stemmed from the assay itself.

In this review, we collated and analysed discriminating dose data from the Liverpool Insecticide Testing Establishment (LITE) and Vector Biology Department at the Liverpool School of Tropical Medicine (LSTM). The *Anopheles* mosquito colonies maintained by each group are profiled at least annually using standard WHO susceptibility tests. Additionally, each group applies deltamethrin selection to resistant strains every 3–5 generations to maintain pyrethroid resistance (i.e., mosquitoes are exposed to deltamethrin using standardised procedures, and survivors are used to maintain the colony). In most instances, this selection follows the same protocol as the WHO susceptibility test (exposure to 0.05% deltamethrin for 1 h). Additionally, when testing novel or repurposed chemistries, a positive pyrethroid control is often used in experiments—overall, these studies represent a set of repeated bioassay (tube or bottle) measurements under uniform testing conditions and using the same mosquito colonies.

For each mosquito strain, the mortality data for profiling, selection, or other experiments were compiled. When colonies of a strain were held in both LITE and LSTM, these data were considered separately. For each strain/insecticide combination, summary statistics of mortality were calculated (the range, interquartile range, mean, median, variance, and standard deviation). A Welch’s *t*-test was used to compare mean mosquito mortality following exposure to different pyrethroids, or the same pyrethroid in different assays. The analysis was conducted with R statistical software version 3.6.2 (12 December 2019) [14].

In general, following exposure of characterised lab strains in WHO tube bioassays under controlled conditions, the level of variability in mortality among test replicates exposed to a single compound was greater in moderately resistant strains (mean mortality > 15%) (Figure 4). In this example, the standard deviation for FUMOZ-R (mean mortality 25.24%) following exposure to permethrin was 29.12, and for Tiassalé 13 (mean mortality 15.97%) it was 16.01. In instances in which insecticide/strain replicate numbers were lower (<10 replicates in some cases), variability was lower (Appendix A and Appendix A). Boxplots summarising mortality of all strains examined to all insecticides can be found in Appendix A.

In CDC bottle assays (with 0.00125 µg/bottle permethrin or 0.00125 µg/bottle permethrin + 400 µg/bottle piperonyl butoxide (PBO) simultaneously [15]), greater variability in mortality was again observed in moderately resistant strains (mean mortality 15–80%) compared to highly resistant (mean mortality < 15%) or more susceptible strains (mean mortality > 80%) (Figure 5). This was mirrored in the PBO treatments in which mortality was greatly increased in the resistant strains and the variability generally decreased (though there was still considerable heterogeneity in the more resistant strains). Further investigation is required to establish the inherent variability in PBO synergism assays, relative to DD bioassays.

When comparing the two testing methods in resistant strains, both mean mortality and variability (standard deviation) in mortality were greater in the CDC bottle bioassay compared to the WHO tube test (in response to their respective discriminating doses), but comparable in susceptible strains (in which almost all mosquitoes died; Figure 6). Previous studies have reported variability in comparability between WHO and CDC bioassay results [16], suggesting reasonable interchangeability in identifying susceptible populations, but less so when substantial resistance is present [7]. Dose-response experiments are perhaps more easily performed using bottles, but very high concentrations may prove difficult due to issues with solubility or crystallization of active ingredients [17].

## 5. Is There Evidence for Divergent Resistance in Lab Colonies Routinely Selected Using a Single Pyrethroid?

Comparing mortality in the same laboratory populations routinely tested against DDs of different pyrethroids allowed for a comparison of whether susceptibility within any of the strains differed between compounds, and whether this changed over time (albeit an imperfect comparison, given the uncertainty around the DDs themselves and the level of variability in assay results already described). In general, susceptibility to permethrin, deltamethrin, and α-cypermethrin was similar within individual strains (Figure 7).

The Tiassalé 13 strain maintained by LSTM exhibited, on average, higher mortality against deltamethrin than to other pyrethroids, though no difference was seen in the Tiassalé 13 strain in experiments conducted at LITE. The differences in mortality between the insecticides was less than that seen in repeated experiments within the same strain (Figure 7), making it difficult to conclude if there were true differences in susceptibility between different pyrethroids between strains based on these data. The laboratory strains tested in this dataset had been selected with deltamethrin for up to 6 years [18]. In the absence of selection against all pyrethroids, we could expect divergence over time if differences existed between the pyrethroids, yet there was no obvious trend towards increasing relative resistance to deltamethrin. When mortality in the WHO tube bioassay was plotted over time, no obvious trends or changes in mortality from year to year across all strains were detected, and any temporal changes within strains seemed consistent across all pyrethroids tested (Appendix A). These data were collected passively during routine monitoring of the resistance profiles of reference colonies, and a targeted investigation into the effects of selection pressure on differential resistance to individual pyrethroids is needed to reach a more robust conclusion.

## 6. What Are Potential Sources of (Non-Resistance-Associated) Variability in the Discriminating Dose Bioassay?

The WHO [4] gives precise parameters for some of the key environmental conditions that should be established when carrying out bioassays. Poor larval rearing conditions (e.g., crowding and/or low food) can have extreme effects on bioassay results [19], but these are relatively easy to control under standard insectary conditions. Nevertheless, details of the rearing conditions employed are often scant in reports of bioassay data, and expanded descriptions would help to assess whether this may be an important source of variability. Time-of-day effects on bioassay results do not seem to be well-explored in the literature, but circadian rhythmicity of many detoxification genes suggest that mosquitoes tested at night may not show the same resistance patterns as those tested during the day [20,21]. This may be significant for the operational interpretation of results, considering that African *Anopheles* typically bite at night. However, this is unlikely to be a major source of variability affecting bioassay data, since tests are typically performed during daytime hours. Nevertheless, reporting of testing times along with bioassay results would be a good practice to adopt more widely.

In contrast, under field conditions, WHO-specified temperature and relative humidity are often difficult to achieve and maintain, and the effects of variation can be highly significant. As part of a genomewide association study with sampling and testing conducted in a field insectary in Uganda that lacked environmental controls, Weetman et al. (2018) detected a strong and highly statistically significant decline in *An. gambiae* mortality as humidity increased (Figure 8A). In this study, temperature also varied, but did not independently account for the statistically significant variation in mortality. In the WHO-IIR (impacts of insecticide resistance multicentre trial), temporal repeatability of results from sentinel sites in Sudan was poor [22], and a significant contributory factor may have been variability in temperature and relative humidity, which correlated strongly (Figure 8B). As temperature and humidity decreased, mortality increased at the discriminating dose in pyrethroid bioassays with the *An. arabiensis* tested. Interestingly, this was the opposite directionality to that observed in Ugandan *An. gambiae*, and may reflect the differences in aridity tolerance between the species [23]. Significant, but inconsistent, effects of temperature on bioassay mortality have also been reported among laboratory colonies of *An. stephensi* [24], *An. arabiensis,* and *An. funestus* [25]. Whether or not the contrast in the direction of effects of humidity and temperature between studies reflects differential physiological adaptations of the species studied, such variability highlights the difficulty in predicting and statistically controlling for temperature and humidity effects. Indeed, these may depend quantitatively on the humidity–temperature optimum-tolerance profiles of the population tested. Nevertheless, studies should record and report these variables accurately, so that caveats can be applied when concluding datasets obtained under differing ambient conditions.

The age of mosquitoes tested is also an important consideration, and multiple studies have shown that mortality in pyrethroid bioassays performed on *An. gambiae*, *An. coluzzii,* and *An. arabiensis* increases with mosquito age [27,28,29,30,31]. However, this pattern may not be universally true across insecticides and resistance mechanisms. Recent work on pirimiphos-methyl-resistant *An. gambiae* from Ghana, in which resistance is strongly determined by combinations of target site mutations, showed no differential trend in mortality over ages spanning 3–15 days [32]. However, provided mosquitoes were reared in the laboratory from larvae or eggs, we are not aware of any results from bioassays that showed decreases in mortality with age. With a preference to test the least-susceptible age group in insecticide bioassays, this argues for the current approach of targeting young (but at least 2-day-old) adults.

Physiological conditions of females, not directly related to age, may play a less predictable role in variation in bioassay mortality. The effect of blood feeding has been primarily studied in laboratory strains of *An. arabiensis* or *An. funestus*, in which a moderate and transient reduction in permethrin and deltamethrin mortality after a single blood meal was detected [33,34]. These findings have recently been replicated with field-collected samples of *An. gambiae* from Kenya [31]. The proposed mechanism for this is the upregulation of a vast number of detoxification genes in response to the oxidative stress caused by the intake of blood by female mosquitoes [35]. The magnitude of effect appeared to be dramatically greater if multiple blood meals were taken before insecticide exposure (up to 60% reduction in mortality for permethrin and deltamethrin, even in 21-day-old females) [34]. Further studies on the same strains showed that multiple bloodmeals appeared to be linked to a sustained enhancement in the ability to defend against oxidative stress, a common toxic effect of pyrethroid exposure [36]. Repeated sublethal prior insecticide exposures might have a similar effect, but results to date are inconclusive [27], possibly because of the conflicting effects of priming via enzyme induction from insecticide pre-exposure, and delayed effects of sublethal exposures on mortality [37]. In the absence of additional studies, the ubiquity and magnitude of the effects of repeated sublethal insecticide exposure, and more concerningly, repeated blood feeding, are difficult to predict, but suggest that in combination with the more estimable age effects, performing bioassays on adult females caught directly from the wild may provide highly variable or even biased results.

A common feature of most published works describing bioassay data is a relatively poor description of the sampling methodology, which is usually performed following an opportunistic plan. Generally, few details are provided to describe the range of collection sites, and often only a single GPS location is given, which can probably be assumed to represent an approximate central point for sites contained within a polygon of unknown size [38]. For comparative studies involving bioassay data, this is problematic because: (a) chances of repeatability are lowered by lack of collection detail, and (b) samples may lack independence as biological replicates, which may introduce bias or inflate statistical power. A priori, the predicted magnitude of this effect is expected to depend on the collection method employed. If adults are collected, they may be either tested directly (noting the inherent problems with testing adults with unknown variation in physiological status and age described above) or used to obtain eggs, which may be combined and reared for adult bioassays. Collected adults would typically be assumed not to be closely related; whilst if their eggs are used, the level of relatedness in the resultant sample would be expected to be roughly proportional to the number of families combined (assuming equal contributions from each). However, for the *An. gambiae* complex, the most common method of obtaining samples involves collecting larvae from larval habitats, presenting a potentially significant, but unknown, likelihood of sampling siblings. A strategy of collecting from as many local larval habitats as possible might reasonably be expected to ameliorate this problem to some extent. Yet, to our knowledge, there has been no previous study examining relatedness levels in collections made following any of the above collection strategies. As part of genomewide association studies using bioassay-based insecticide-resistance phenotypes, larval samples were collected from Yaoundé, Cameroon, and Dodowa, Ghana in 2006 [39], and adults from Tororo, Uganda in 2008, from which offspring were obtained for bioassay testing [23]. Further samples were obtained from recently and long-established colonies at LSTM, and all samples were genotyped using a custom Illumina array. More recent collections were made from over 50 locations (each represented by several larval habitats within a radius of a maximum of a few kilometres, and often much less) across southern Ghana in 2016. Genomes of a random sample from each collection were sequenced at low coverage [32]. In each dataset, relatedness categories among the samples were estimated (Figure 9).

Results proved to be surprising. Larval collections in 2006 contained only approximately 5% of siblings, and those from within the same locations (i.e., sets of local larval habitats) in the 2016 collections showed a similar overall average, though occasionally sites showed much higher values (maximum = 46% related as half- or full siblings; see Appendix A). This suggests that relatedness within larval habitats is much lower than might typically be assumed, and samples dominated by siblings are probably the exception, rather than the norm, provided efforts are made to sample as many locally accessible sites as possible. This is concordant with recent results from *An. arabiensis* showing that productive larval habitats contained many larvae because they contained many families, rather than large numbers from single or few families [42]. Relatedness among the adults collected from houses in Uganda was similar to that among the larval collections, and, as expected, all the estimates from field sampling contrasted very markedly with the majority of close relatives seen in the recently established colony, and especially the long-established Kisumu strain. Overall, these results suggested that with reasonable diligence, most larval samples of *An. gambiae* might be assumed as broadly unrelated in locations where multiple larval habitats are available, providing little problem with the assumptions of independence for statistical models. When obtaining larvae is difficult, obtaining eggs from many females presents a reasonable alternative, including reporting details of the number of egg batches combined alongside data.

A final consideration in sampling is species identification. Failure to differentiate morphologically cryptic species within complexes or groups can create significant biases when comparing results among studies. When relative species composition varies in space or time, failure to identify which are being tested can lead to misinterpretation of causality if insecticide resistance differs interspecifically, which is often the case for *An. gambiae* or *An. coluzzii* vs. other species complex members [43]. Multiple, cheap, and reliable molecular assays are available to identify species, and their application is crucial, though initial morphological identification to the level of species complex or group is always strongly advised [44].

## 7. What Is the Evidence for the Existence of Divergent Resistance between Pyrethroids? Can Differences Seen in Molecular Studies (Section 3) Be Detected in Wild Mosquito Populations?

The WHO intensity bioassay [4] is likely to have a lower measurement error than the WHO discriminating-dose bioassay, as each assay combines 3–6 repetitions across different insecticide intensities. A dataset was analysed that contained intensity bioassay results from the Presidents Malaria Initiative, WHO Malaria Threats database [45], and studies collated by Moyes et al. [38]. There were insufficient data available that directly compared different Type I and Type II pyrethroids in the same experiment. To overcome this, data were pooled at the country level to compare studies that tested permethrin and deltamethrin. This dataset consisted of 4745 individual mortality estimates from 1583 intensity bioassays across 18 countries in sub-Saharan Africa. Data from CDC bottle bioassays and WHO tube assays were analysed separately to investigate pyrethroid-specific resistance in wild mosquito populations. The analysis was restricted to the African continent due to the availability of data.

A Bayesian binomial model was developed to generate dose-response curves from raw intensity bioassay data. Separate curves were originally fit for each insecticide to the whole dataset to illustrate overall trends (Figure 10). Separate models were then fit to each set of concentrations to estimate the individual median lethal concentration (Lc_50_; i.e., the concentration at which 50% of mosquitoes tested died). Mean LC_50_ estimates by country and year were calculated from individual estimates, with 95% credible intervals generated using bootstrapping methodology.

Overall, across all data the dose-response curves were similar. On average, at an exposure of up to 10× the DD, the best-fit curve indicated that mortality induced by deltamethrin was higher than that of permethrin (Figure 10). This overall consistently shaped dose-response curve is compatible with the hypothesis that the two insecticides have different DDs. If the concentration of permethrin originally selected as “discriminating” induces a higher level of mortality than that selected for deltamethrin, then this discrepancy will be propagated across all concentrations (as they are relative to the DD at 2×, 5×, and 10×). The combined curves suggested that this is reversed when extrapolated to higher intensities, though this is likely an artefact of the shape of the curve used to describe the dose–response relationship combined with relatively few data points above 10× concentrations, though this needs to be verified.

When analysing data at the country level, different trends were observed in different countries (Figure 11). High LC_50_ variability was seen in all locations, with substantially greater differences seen within-country than between countries. On average, differences between insecticides appeared marginal, with many overlapping mean LC_50_ estimates across both insecticides. Lower LC_50_ estimates (higher mortality) were seen for deltamethrin across most countries and assay types. There were several countries where this trend was, on average, reversed (i.e., higher LC_50_ estimates for deltamethrin in Burkina Faso and Cote d’Ivoire for WHO tube assays). Nevertheless, the difference between the LC_50_ estimates of the two insecticides was substantially less than the differences seen between assays of the same insecticide conducted in the same country, suggesting high variability but no clear pattern.

If there were differences in the suite of resistance mechanisms against Type I and Type II pyrethroids, and these mechanisms were established in populations, then it might be expected that resistance would diverge over time if selection pressures were continual. Selection is thought to be driven, at least in part, by ITN use, so this selection pressure is likely to be relatively consistent, as ITNs are typically replaced every three years. Differences in time were difficult to discern from these resistance-intensity data, as results were only available for 1–5 years. Nevertheless, the results were surprisingly consistent over the different years, with those countries showing differences between pyrethroids generally persisting (Figure 12). On average, differences in mortality between insecticides did not increase over time, providing support for the cross-resistance hypothesis.

Importantly, the difference between insecticides was likely not substantial enough to have a meaningful public health impact. The absolute difference in mortality at the DD dose predicted by the dose-response model was relatively low, varying from 2–27% between countries with multiple years of data (data not shown). Temporal trends, when they did appear, were also relatively minor, changing on average by only a small percentage over the timeframe. Evidence from the CDC bottle assay in Mali consistently showed higher mortality after deltamethrin rather than permethrin exposure, which remained constant over multiple years (Figure 11). However, a negligible difference was seen in the WHO tube assay from the same region (Figure 11), so it is unclear whether this may be due to a sampling/procedural artefact or differences within the country.

Whilst intensity bioassays may help in decreasing measurement errors compared to DD bioassays, the phenotypic field data remains very noisy. Whether these differences represent true variability in the local mosquito populations or are an artefact of the assay is unclear. Overall, these data indicated there was no consistent difference in mortality between deltamethrin and permethrin.

## 8. Do Mosquitoes, Resistant or Susceptible, Exhibit Different Behavioural Responses to Different Pyrethroids?

Vector populations can respond to IRS or ITN selection pressure with changes in behaviour, such as shifts in time or location of biting, resting site preferences, or host preference to avoid encountering the insecticide. However, behavioural resistance may have many other less apparent forms; e.g., changes in sensitivity to repellent or irritant properties, or modified blood-feeding behaviours. Potentially, less detectable changes might be associated with highly visible secondary consequences; e.g., a thicker cuticle due to resistance could result in changes in flight behaviour. If there are differences in behavioural responses to different pyrethroids, then behavioural resistance might diverge such that resistance to one pyrethroid might be overcome by deploying a different one, or deployment choice might need to consider whether certain pyrethroids are more or less likely to drive behaviour in resistant mosquitoes that could lower an ITN’s efficacy.

The mechanisms of insecticide resistance in malaria vectors are being studied and characterised extensively at the molecular level. Yet, knowledge of behavioural change associated with resistance is relatively poor, and few studies have directly compared the behavioural response to different pyrethroids within the same study. Before the emergence of resistance, an early hut trial in The Gambia concluded that permethrin was the most repellent pyrethroid, followed by λ-cyhalothrin, deltamethrin, and lastly cypermethrin [46]. Over a decade later, Hougard et al. [47] tested bednets with various combinations of bifenthrin and carbosulfan against both resistant and susceptible *An. gambiae* s.l., and reported no differences in entry rates between treatments or vector populations. Asidi et al. [48] tested bednets treated with α-cypermethrin, λ-cyhalothrin, permethrin, deltamethrin, or carbosulfan against resistant *An. gambiae* s.l. in Côte d’Ivoire. Here, all nets performed similarly, with none exhibiting any deterrent effects until they had been washed, after which all treatments reduced entry rates by approximately half. Cooperband and Allan [49] found *An. quadrimaculatus* spent significantly longer times resting on surfaces treated with deltamethrin than with bifenthrin or λ-cyhalothrin, but only after initial contact was made. Hughes, Foster, et al. [50] found no evidence for deterrence in *An. gambiae* s.l, but recorded lag times between first net contact and the start of blood feeding of 1 min with untreated nets, and 2.5 and 3 min for Olyset and PermaNet 2.0 nets, respectively. Other studies have described the behavioural responses of mosquitoes to pyrethroid-treated nets. However, there was great variability in the study designs, behavioural definitions, net treatments, and mosquito species reported in the literature. Studies investigated numerous diverse wild vector populations at different locations, on different dates, with very different or uncharacterised levels of resistance. Consequently, the results were highly variable, with little indication of a conclusive trend among the behavioural responses elicited by individual pyrethroids, let alone anything to distinguish behaviours unique to different insecticides within the pyrethroid class.

## 9. How Suitable Are Existing Resistance-Monitoring Methods for the Detection or Measurement of Behavioural Resistance?

Since the impact of any insecticide-based control method is determined ultimately by the mosquitoes’ behavioural response at or near the interface of insecticide delivery, the selection of ITNs should ideally be based on evidence derived from appropriate assays that capture the range of behaviours that influence the ITN’s performance. The discriminating-dose and resistance-intensity bioassays (whether WHO tube or CDC bottle) currently used to monitor resistance were not designed to allow for or monitor behavioural variation.

Bioassays such as the cone and tunnel tests record knockdown or mortality (and blood-feeding rate in tunnel tests) of young adult female mosquitoes following unnaturally high levels of exposure to an active ingredient under highly artificial conditions; i.e., forced, without the presence of a host, or using a non-natural animal host. Measuring the efficacy of insecticides in such an environment will not predict how the eventual insecticidal net products or residual spray preparations will perform under field conditions, hampering informed deployment decisions. Similarly, a change in behaviour in a mosquito exposed to a pyrethroid might well confound the results of such bioassays. For example, an increase in sensitivity could reduce contact times at the treated surface, resulting in lower mortality, whereas an increase in tolerance could increase the contact time and mask existing resistance.

Discriminating-dose assays (WHO tube and CDC bottle) were intended as a litmus test for the emergence of resistance in mosquito populations to evaluate fast-acting pyrethroids before the extent of resistance seen in Africa today became established. They were not designed to measure quantitative differences between mortality rates to inform product choices, and certainly not to draw any comparisons between the efficacy of different products. Their limitation is their inability to capture the full range of possible behavioural and sublethal effects—such as impacts on longevity, reproductive output, or development of *Plasmodium* spp. Improving, augmenting, or replacing tests as affordable, rapid, and simple as the existing WHO tests with new assays that retain those properties, as well as adding the ability to capture, distinguish, and measure a range of outcomes without ambiguity, will be a challenge.

Carrasco et al. [51] attempted to capture all anticipated behavioural events and other potential outcomes following insecticide vector control within a framework to guide classification and investigation. A better understanding of behavioural responses, to both insecticides and specific products, and how they differ between susceptible and resistant mosquitoes should inform the deployment of the most effective products. Ideally, given the vast range of behaviours that could be impacted and might need to be quantified, resistance-monitoring efforts should focus methods to detect changes in those behaviours most likely to affect a product’s performance. To meet this need, there have been a number of advances in development of novel methodologies [50,52,53,54] to collect the essential data about the behavioural responses of *Anopheles* under more operationally relevant conditions, including large-scale testing arenas [55,56].

## 10. Discussion

### 10.1. The Evidence for Divergent Resistance within the Pyrethroid Class

Molecular analysis of metabolic resistance, together with analysis of phenotypic resistance in mosquito populations (including analyses of intensity data, of diagnostic dose bioassay data from populations that have been tested with multiple pyrethroids, and of spatiotemporal trends), provide evidence that there is strong cross-resistance among pyrethroids, particularly between permethrin and deltamethrin. P450s SAR (structure–activity relationship) findings concluded that the more commonly used pyrethroids examined were the most vulnerable to metabolic attack (by cytochrome P450s), while bifenthrin, λ-cyhalothrin, and α-cypermethrin were less vulnerable to metabolic attack. Bioassay data from *Aedes aegypti* and *An. sinensis* suggested that bifenthrin may demonstrate relatively low cross-resistance with other more commonly used pyrethroids. Bifenthrin has not been widely used in malaria control in Africa and no discriminating dose has been defined, but its potential use in malaria vector control warrants further investigation. There is also evidence that resistance to etofenprox could diverge from resistance to the more commonly used pyrethroids; however, further investigation regarding vulnerability to metabolic attack by P450s is required.

In field populations, variability in discriminating-dose and dose-response assay mortality was high. This variability was predominantly at a fine geographical scale (i.e., assays done within 50 km of each other were highly variable), indicating that if there were a difference between Type I and II pyrethroids, it would be very local and beneath the size of the regions to which insecticidal nets are currently allocated. There was good evidence that the mortalities from exposure to deltamethrin, permethrin, α-cypermethrin, and λ-cyhalothrin were strongly correlated across *An. gambiae* s.l. populations. These correlations were also seen for deltamethrin, permethrin, and λ-cyhalothrin (α-cypermethrin was not tested) in the *An. funestus* subgroup, and in all three of the main malaria vectors within the *An. gambiae* complex.

### 10.2. The Suitability of Current Testing Methods to Monitor Insecticide Resistance and Make Vector-Control Decisions

Deployment decisions for ITNs are being guided by information arising from the discriminating-dose and resistance-intensity bioassays, but it is not clear how well differential mortality in WHO tube or CDC bottle bioassays predict how well an ITN treated with one or another pyrethroid will perform in a specific site. Bioavailability may play an important role in the relative efficacy of different ITNs, and testing for the relative performance of different nets against field populations would provide more directly relevant information for deployment decisions, alongside or in place of conventional bioassays. Given the limited products available for vector control, as well as narrow collection of available chemistries, programmes must make ITN deployment decisions based on the data that can realistically be collected. The current monitoring system for insecticide resistance is imperfect and should be adapted to make better use of the available resources, while being mindful that limited mosquito collections preclude the testing of all insecticides/products.

In general, following exposure of characterised lab strains in WHO tube bioassays under controlled conditions, intrastrain mortality to permethrin, deltamethrin, and α-cypermethrin were similar. However, in intermediately resistant strains, some divergence in mortality rates was observed. However, importantly, the level of variability in observations of mortality between tubes (measured using standard deviations) was also greater in these intermediately resistant strains, which reduced certainty in apparent contrasts between insecticides. Discriminating-dose assays are poor tools for quantitative analysis of resistance levels where resistance is established, producing the most variability in results in laboratory colonies and field populations where resistance was moderate and mortality was intermediate, which is likely to be the case for all or most pyrethroids in most populations of African malaria vectors. For comparisons across insecticides, intensity assays (e.g., 1×, 5×, 10×) suffered from the same problem as the discriminating doses on which they depended—an apparent lack of parity across pyrethroid insecticides. Quantitative dose-response assays, which do not depend on a discriminating doses, are recommended for robust comparisons between insecticides.

All bioassays are vulnerable to ambient conditions, including humidity and temperature, in addition to other environmental effects more easily standardized by the user. It is crucial that deviations from optimal conditions are reported, along with the improved provision of sampling details, to understand extrinsic factors that could influence bioassay results.

## 11. Conclusions

Evidence suggests that in areas where pyrethroid resistance exists, different results in insecticide susceptibility assays with specific pyrethroids currently in common use (deltamethrin, permethrin, α-cypermethrin, and λ-cyhalothrin) are not necessarily indicative of an operationally relevant difference in potential performance. Consequently, it is not advisable to use rotation between these pyrethroids as an insecticide resistance management strategy. Less commonly used pyrethroids (bifenthrin and etofenprox) may have sufficiently different modes of action, though further work would be needed to examine how this may apply to insecticide resistance management.

## Figures and Tables

**Figure 1 insects-12-00826-f001:**
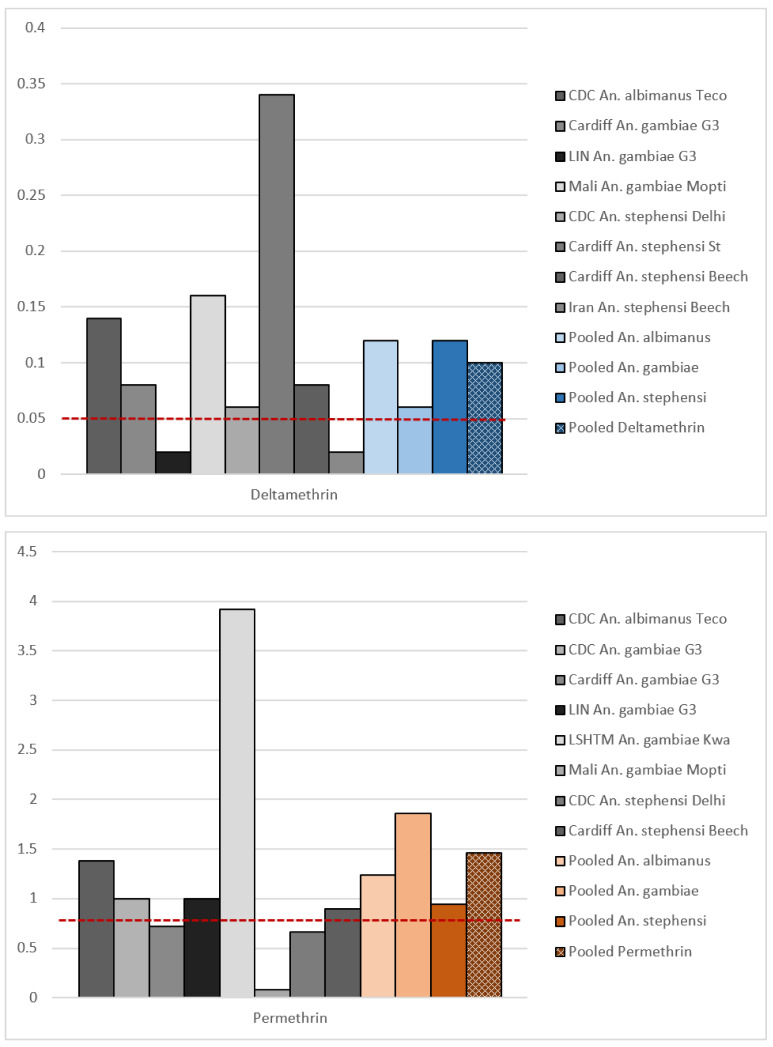
Calculated discriminating doses (%) for deltamethrin (**top**) and permethrin (**bottom**). Points show individual site/strain combinations, and data pooled by species or overall and by the insecticide. Site/strain combinations testing <3 concentrations of an insecticide and datasets that were not robust enough to calculate lethal dose matrixes were excluded. Discriminating doses were set at 2× the calculated lethal dose at which 99% (LD_99_) of test mosquitoes were killed. The dashed red line represents the current WHO-recommended DD (0.05% deltamethrin and 0.75% permethrin). LSHTM *An. stephensi* data are omitted here to improve visualisation of other data points, as their calculated DDs were high. A version with this included can be found in Appendix A. Bar charts displaying mosquito mortality (%) following exposure to permethrin and deltamethrin in individual WHO tube bioassays can be found in the Appendix A.

**Figure 2 insects-12-00826-f002:**
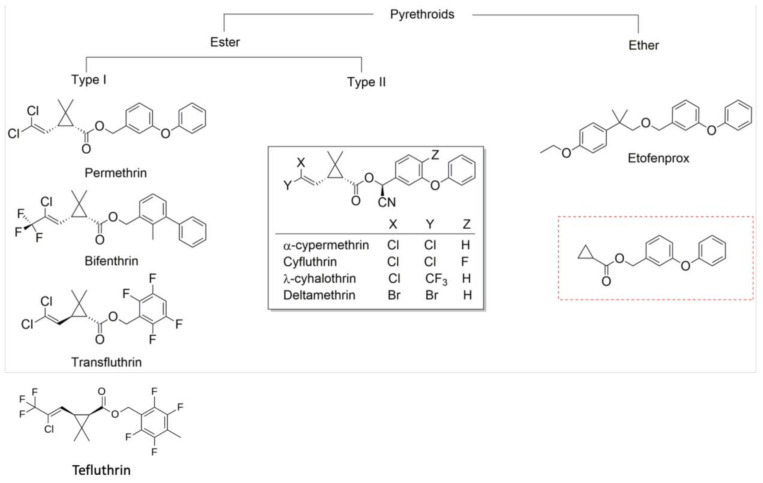
Chemical structure of pyrethroid insecticides used for malaria vector control. The common scaffold of pyrethroids, boxed in red, was identified by searching 230 million compounds available in the ZINC database (https://zinc.docking.org, accessed on 23 February 2020). Adapted from [12].

**Figure 3 insects-12-00826-f003:**
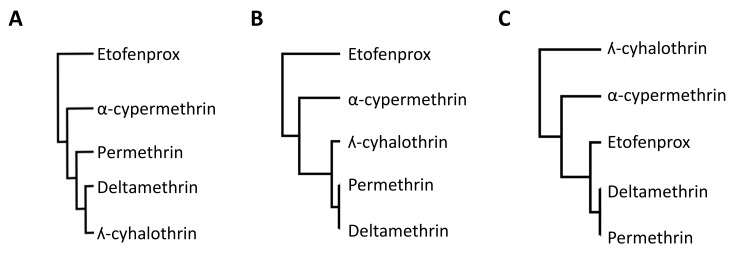
Hierarchical relationships among pyrethroids defined using data on resistance in vectors and functional activity data. The dendrograms were constructed using correlations in mortality across African malaria vector populations (Pearson’s correlation coefficient) (**A**), binding affinity values (IC50), (**B**) and insecticide depletion values (%) (**C**). Adapted from [12].

**Figure 4 insects-12-00826-f004:**
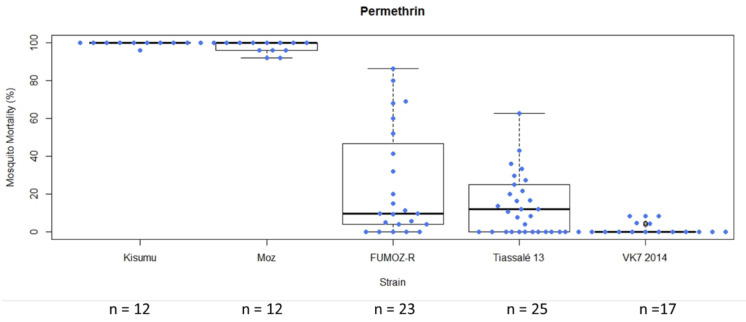
Box plot summarising mosquito mortality following exposure to permethrin 0.75% in a standard WHO tube bioassay in LITE strains. Each box represents a different mosquito strain. Coloured circles and n values indicate each tube replicate (not total mosquito numbers).

**Figure 5 insects-12-00826-f005:**
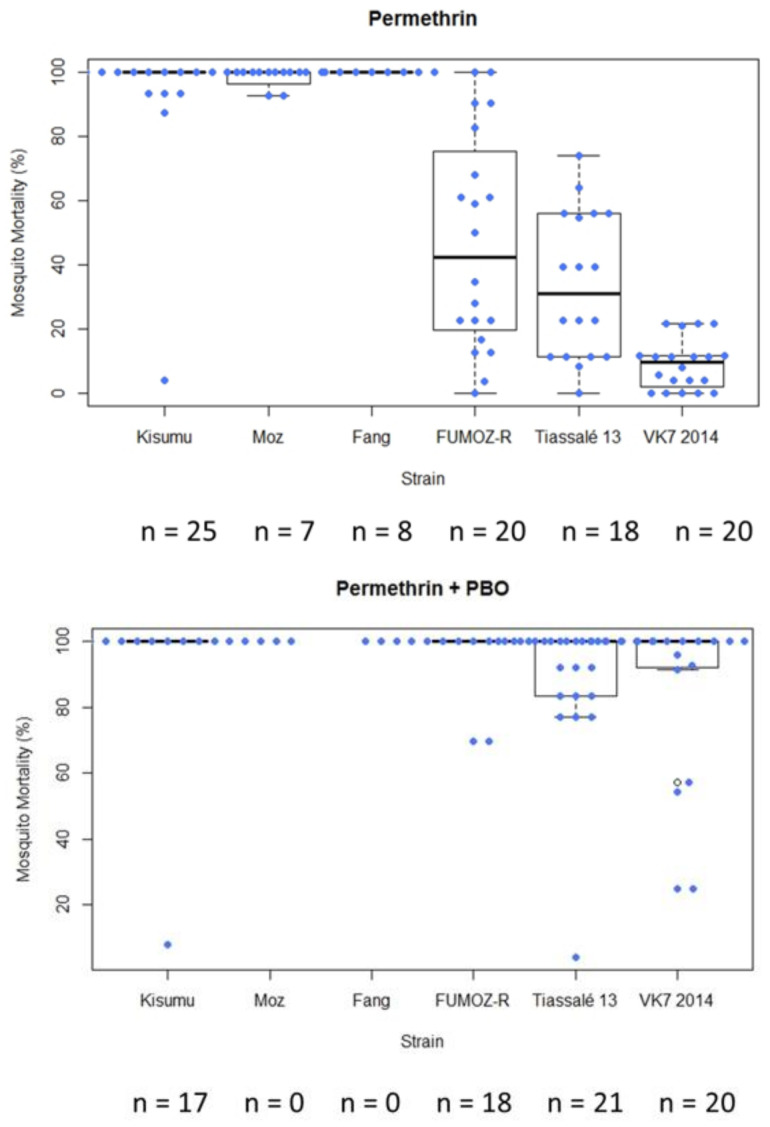
Box plot summarising mosquito mortality following exposure to permethrin 20 μg /bottle (**top**), or permethrin 20 μg/bottle + PBO (**bottom**) in a standard CDC bottle bioassay. Each box represents a different mosquito strain. Strains here are those maintained by LITE at LSTM. Coloured circles and n values indicate each replicate bottle.

**Figure 6 insects-12-00826-f006:**
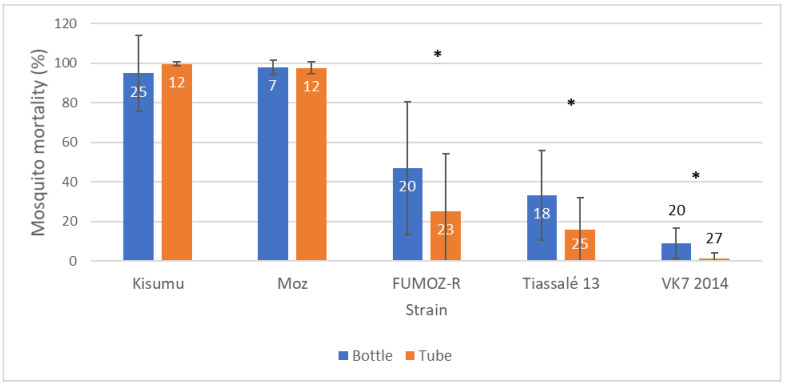
Average mosquito mortality following exposure to discriminating doses of permethrin in CDC bottles (0.00125 µg/bottle) (blue bars) or WHO tube (0.75%) bioassays (orange bars) in LITE strains. Numbers above or in bars indicate the number of replicate bottles or tubes. Error bars show the standard deviation to indicate variability between replicates. Asterisks above bars indicate where mean mortalities were significantly different (*p* < 0.05, Welch’s *t*-test).

**Figure 7 insects-12-00826-f007:**
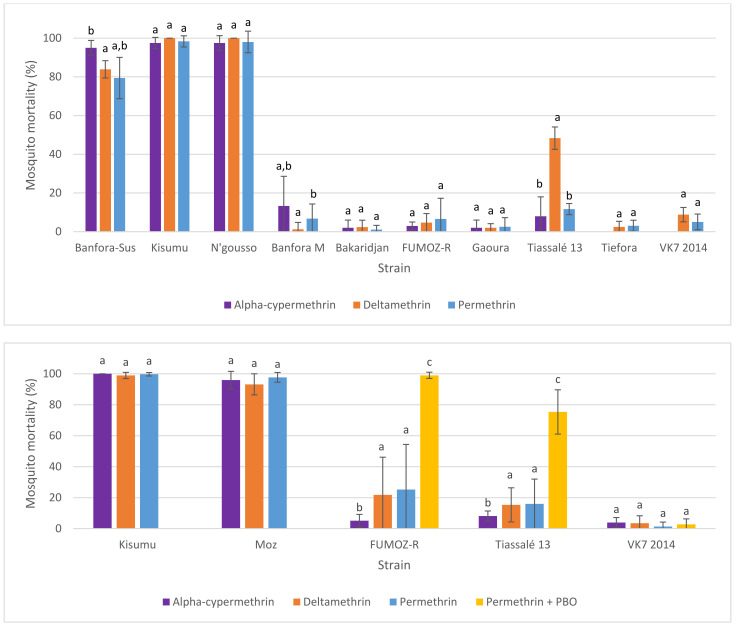
Average mosquito mortality following exposure to α-cypermethrin 0.05% (purple), deltamethrin 0.05% (orange), permethrin 0.75% (blue), or permethrin 0.75% preceded by piperonyl butoxide (PBO) (yellow) in a standard WHO tube bioassay in LSTM (**top**) or LITE (**bottom**) strains. Bars sharing the same superscript letter were not significantly different (*p* < 0.05, Welch’s *t*-test). Error bars show the standard deviation to indicate variability between replicates. The *p*-values are shown in Appendix A.

**Figure 8 insects-12-00826-f008:**
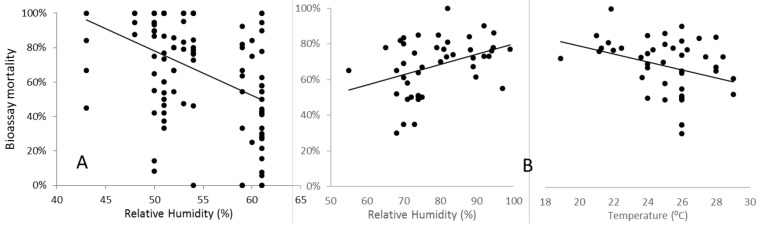
Effects of environmental conditions recorded in field insectaries during the insecticide exposure period on bioassay mortality (**A**) humidity on permethrin assays performed on Ugandan *An. gambiae* [26] and (**B**) humidity and temperature on deltamethrin assays on Sudanese *An. arabiensis* [22]. All regression lines were highly significant.

**Figure 9 insects-12-00826-f009:**
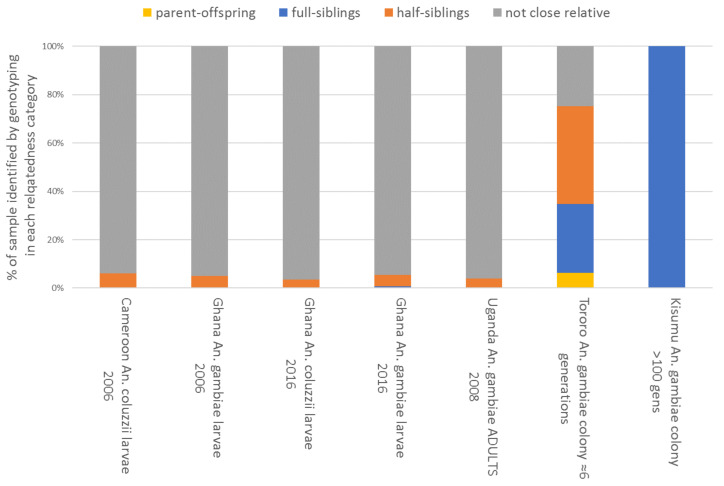
Genotype-based identification of close relatives in female samples collected as larvae from nearby collection locations in 2006 and 2016, and as adults in 2008; samples from both a recently established and a very long term colony are also shown for comparison [40]. Results from 2016 were estimated from data on ≈160 samples for each species at 2229 chromosome 3 SNP markers (see Appendix A). All other data were from 286 chromosomes 3 SNP markers (37) with field sample sizes of ≈180 for adults and 600–700 for larvae. Relationship categories were estimated using ML-Relate [41].

**Figure 10 insects-12-00826-f010:**
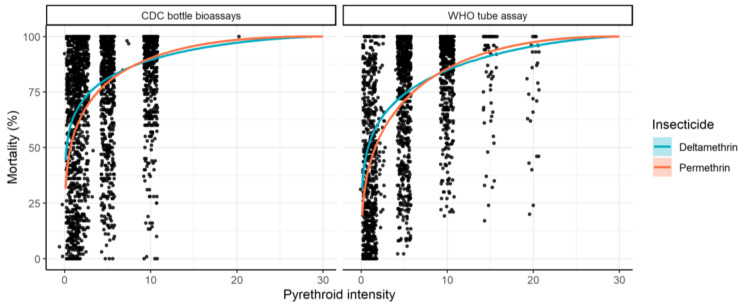
Overview of the intensity bioassay data used for analysis: points represent raw mortality data at each respective insecticide intensity with the modelled dose-response curve shown by Table 1 at 1×, 2×, 5×, 10×, 15×, or 20×.

**Figure 11 insects-12-00826-f011:**
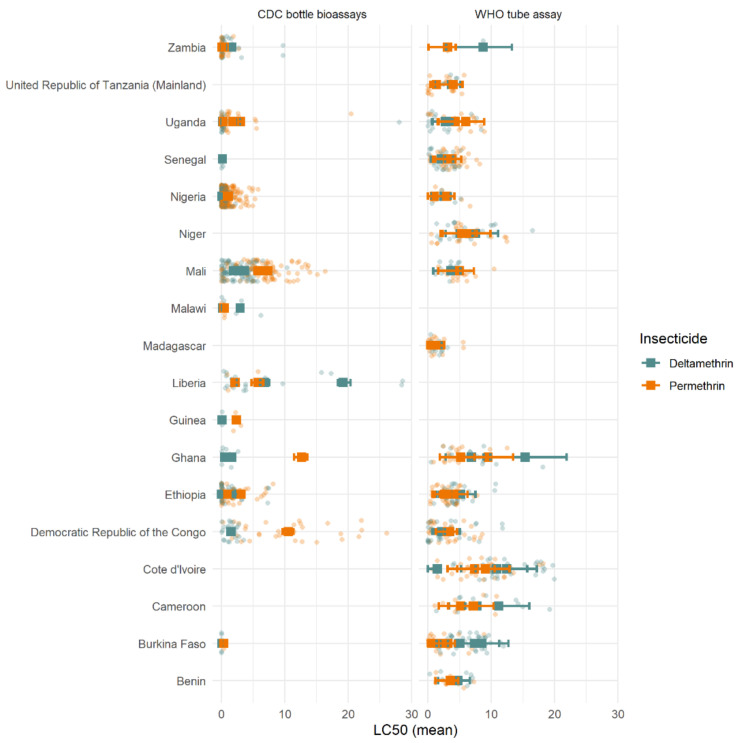
Estimated LC_50_ values for each country and assay type. The solid squares represent the mean value per country with each individual intensity bioassay LC_50_ value shown as the light points. Horizontal coloured lines indicate 95% credible intervals of the mean LC_50_ estimates. The x-axis units are DD concentration 10×, 20×, and 30×.

**Figure 12 insects-12-00826-f012:**
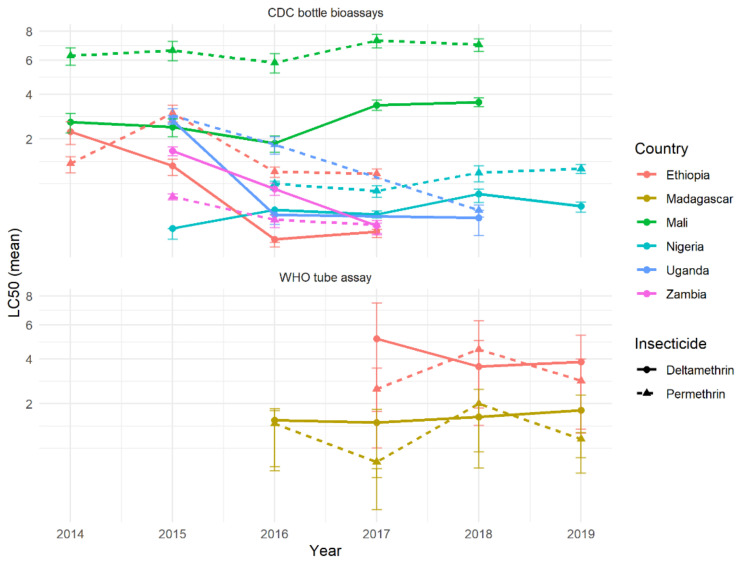
Country-specific mortality trends over time for countries with data from 3 or more consecutive years for both insecticides (per assay type). Mean LC_50_ values for each insecticide per country are shown for each assay and year. Different colours represent different countries, and each insecticide is shown by point and line type (permethrin in triangles and dashed lines, and deltamethrin in point and solid lines). Ninety-five credible interval estimates for the mean LC_50_ estimates are shown with the vertical whiskers, whilst horizontal lines link countries (though sites varied within countries over time).

**Table 1 insects-12-00826-t001:** Probit analysis of 1998 WHO multicentre study. The discriminating dose is 2× the LD_99_. Abbreviations: LD = lethal dose; CI = confidence interval; DD = discriminating dose; DoF = degrees of freedom.

	Deltamethrin	Permethrin
Number exposed	8258	9582
LD_99_ (95% CI)	0.05 (0.023–1.166)	0.73 (0.452–2.067)
Calculated DD	0.1	1.46
Chi-square ^1^	2215.41	2055.49
Heterogeneity (DoF)	39.56 (56)	32.12 (64)
Current WHO DD	0.05	0.75

^1^ Chi-square provides a measure of fit.

## Data Availability

All data generated or analyzed during this study are included in this published article.

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
