# Peer review of "(untitled)"

_insects, 2021, doi:10.3390/insects12090826_

Round 1

Reviewer 1 Report

This is a timely and well-written review. I was surprised to see the differences in DDs between the present study and the 1998 multicenter study (analyzing the same dataset, questioning the comparability of permethrin and deltamethrin data), and learned a great deal about pyrethroids and insecticide susceptibility test procedures. The suggestions on how to improve sample/data collection, to improve the usefulness of the collected data, was an eye-opener. I would strongly suggest to capture all those recommendations (and ideally the recommended units) in a separate box, as this would be very useful for e.g. the WHO, researchers, NMCPs and their partners.

I only have a few other comments:

- I do not understand Figure 3, and the associated text, as there are no details on the methodology (in particular on the functional activity data).

- Figure 6, and the associated text: Which DDs are used in the WHO tube and CDC bottle assays (probably the recommended values, but that would be good to mention), and how fair is it to compare these two tests with their differences in e.g. dose, substrate, final readout?

Lines 286-290. Details on the methodology are missing (the fact that these were WHO tube tests, how climatic variable were measured, if these were only measured during the exposure period or during the subsequent 24h period as well, or maybe they were based on daily means from a nearby weather station?)

- Lines 310-311 (Whether this pattern is universal…) and lines 316-317 (This argues for the current approach) contradict each other.

Additional comments:

Line 129-130, include reference (‘as their cuticles may not have hardened’)

Line 147. ‘Figure 2’

Lines 193 and 206, how would you define ‘moderately’?

Line 220, reference error, I imagine it refers to Figure 6?

Figure 7, there is no ‘A’ or ‘B’, maybe use (top) and (bottom)

Line 402, note that this citation is not a number

Section 10.1: there are no citations. Especially lines 582-583 and lines 583-585 need one

Author Response

The suggestions on how to improve sample/data collection, to improve the usefulness of the collected data, was an eye-opener. I would strongly suggest to capture all those recommendations (and ideally the recommended units) in a separate box, as this would be very useful for e.g. the WHO, researchers, NMCPs and their partners.

  • We thank the reviewer for their interest in this section, and the suggestion of a box of recommendations. Although we agree that this is a subject that warrants expanded discussion in another forum, we think that adding a box of recommendations for just one section would add confusing emphasis in the manuscript. We have therefore decided not to incorporate this suggestion.

- I do not understand Figure 3, and the associated text, as there are no details on the methodology (in particular on the functional activity data).

  • Following text added: To assess cross-resistance within the pyrethroids in terms of their interactions with key cytochrome P450 enzymes (hereafter P450s) and resistance in vector populations, P450 functional activity data with pyrethroids were compared with field mortality data [11]. Figure 3 represents the relationships among pyrethroids in terms of their binding affinity to and depletion by key P450 enzymes known to confer metabolic pyrethroid resistance in Anopheles gambiae (s.l.) in comparison to mortality data among pyrethroids.

- Figure 6, and the associated text: Which DDs are used in the WHO tube and CDC bottle assays (probably the recommended values, but that would be good to mention), and how fair is it to compare these two tests with their differences in e.g. dose, substrate, final readout?

  • Concentrations have been added to graph. The purpose of the graphs is to show difference in variability between the methods when using the discriminating doses selected for each of those methods. We are not seeking to compare results from these methods in any other way.

Lines 286-290. Details on the methodology are missing (the fact that these were WHO tube tests, how climatic variable were measured, if these were only measured during the exposure period or during the subsequent 24h period as well, or maybe they were based on daily means from a nearby weather station?)

  • Texted edited to address these comments (see tracked changes)

- Lines 310-311 (Whether this pattern is universal…) and lines 316-317 (This argues for the current approach) contradict each other.

  • Texted edited to address these comments (see tracked changes)

Additional comments:

Line 129-130, include reference (‘as their cuticles may not have hardened’) – reference added

Line 147. ‘Figure 2’ - edited

Lines 193 and 206, how would you define ‘moderately’? – definitions added (mean mortality 15-80%)

Line 220, reference error, I imagine it refers to Figure 6? – Already refs figure 6.

Figure 7, there is no ‘A’ or ‘B’, maybe use (top) and (bottom) - Updated

Line 402, note that this citation is not a number - Updated

Section 10.1: there are no citations. Especially lines 582-583 and lines 583-585 need one – These lines are sumarising the work conducted in the manuscript, and we therefore do not believe citations are necessary. To make this clearer we have changed the word ‘studies’ to ‘analysis’ in two places (see tracked changes).

Reviewer 2 Report

This manuscript provides an excellent and detailed analysis of the challenges in various aspects of measuring insecticide resistance. The problems that are raised are key questions that have been faced by most researchers in the field and to have such a comprehensive review of all these issues, that are often considered in a fragmented manner is particularly useful. Bar the small editorial commentary on the manuscript itself, the document is largely satisfactory.

Line 147: Figure?

Line 220: This is a problem with referencing programme that must be attended to. 

The authors ask the key question about resistance to the most widely used class of insecticides the pyrethroids. The authors probe the usefulness of discriminating doses by demonstrating practical problems with the determination of these values. This has far-reaching consequences about the definition of insecticide resistance. The authors then consider the effects of insecticide resistance intensity and differential resistance to different classes of pyrethroids. The manuscript is a careful dissection of the current protocols associated with the detection of resistance.  

The topic is original as this metanalysis brings together a range of critical questions in the field of mosquito insecticide resistance that have not been considered together before. The authors ask key questions that have been considered by researchers in the field but pull together the critical questions in a unique manner. The manuscript also highlights a number of gaps in the literature that may not easily have been considered without the “bigger picture” that has been presented in the manuscript.

The discussion in particular highlights the fact that pyrethroid resistance is not uniform, highlighting the divergent resistance to different pyrethroids and which are prone to cross resistance. Furthermore, they give a succinct summary of the flaws with the current resistance testing protocol as well highlighting the importance of the resistance intensity testing, which is an important consideration that is not nearly given as much attention given the critical information this type of testing can give.

The scope of this work is larger and more inclusive than most other studies of the field. The examination of the reported pyrethroid resistance from an extensive range of African countries as well as a range of species is not as intensively covered in a single publication. This publication could provide critical information that can be used to inform choices made by control programmes.

The methodology is suitable.

Appropriate conclusion are drawn from a strong evidence base. The experimental evidence presented is very well suited to answer the question of the suitability of the current techniques to evaluate resistance and the conclusions drawn are both appropriate and salient

The references are appropriate. The data is accurately interpreted and contexualised, and the conclusions from the referenced studies do not speculate outside from the confines of the data. The references also have remarkable depth and covers the range of key findings on a range of subjects related to many aspects of pyrethroid resistance.

Author Response

We thank the reviewer for their positive comments.

Line 147: Figure? - Updated

Line 220: This is a problem with referencing programme that must be attended to - Updated